**Subject Area:**
biochemistry/cellular biology/genetics

rhomboid, pseudoenzyme, membrane protein, trafficking, inflammation, ADAM17

**Author for correspondence:**
Matthew Freeman
e-mail: matthew.freeman@path.ox.ac.uk

# The molecular, cellular and pathophysiological roles of iRhom pseudoproteases

Iqbal Dulloo, Sonia Muliyil and Matthew Freeman

Dunn School of Pathology, University of Oxford, South Parks Road, Oxford OX1 3RE, UK

  ID, 0000-0001-6899-8338; SM, 0000-0003-1374-9304; MF, 0000-0003-0410-5451

iRhom proteins are catalytically inactive relatives of rhomboid intramembrane proteases. There is a rapidly growing body of evidence that these pseudoenzymes have a central function in regulating inflammatory and growth factor signalling and consequent roles in many diseases. iRhom pseudoproteases have evolved new domains from their proteolytic ancestors, which are integral to their modular regulation and functions. Although we cannot yet conclude the full extent of their molecular and cellular mechanisms, there is a clearly emerging theme that they regulate the stability and trafficking of other membrane proteins. In the best understood case, iRhoms act as regulatory cofactors of the ADAM17 protease, controlling its function of shedding cytokines and growth factors. It seems likely that as the involvement of iRhoms in human diseases is increasingly recognized, they will become the focus of pharmaceutical interest, and here we discuss what is known about their molecular mechanisms and relevance in known pathologies.

## 1. Introduction

Rhomboids are intramembrane serine proteases, first discovered in *Drosophila*, where they proteolytically release membrane-tethered EGF ligands, thereby activating signalling [1,2]. Rhomboids were found in a burst of discovery of several different families of intramembrane proteases which collectively introduced the radical idea of regulated proteolysis within membrane lipid bilayers [3]. Successive studies identified and characterized many other rhomboids in multiple organisms [4,5]. While the function and expression of several fly rhomboids is now well documented, the biological roles of the mammalian counterparts are only beginning to be understood.

Beyond the conservation of rhomboid proteases, more extensive bioinformatic analyses identified a much wider rhomboid-like superfamily [5,6] comprising rhomboid proteases, as well as many relatives that contain residues that disrupt their active site, rendering them proteolytically inactive. Among these 'pseudoproteases', the iRhoms, the focus of this review, are the most closely related to the rhomboid proteases. More distant relatives include the Derlins, UBAC2, RHBDD3 and other proteins that have been barely characterized [6,7] (figure 1). Rhomboid-like proteins are localized in many cellular membranes and, despite large gaps in our knowledge of their function, a wide range of roles have already been uncovered, such as intercellular signalling, mitochondrial dynamics, parasite invasion and protein quality control [8,9]. This review will focus on iRhom pseudoproteases and their (i) known physiological client proteins and associated cellular processes, (ii) structure and function, (iii) modes of regulation, and (iv) relevance to human diseases.

royalsocietypublishing.org/journal/rsob Open Biol. 9: 190003

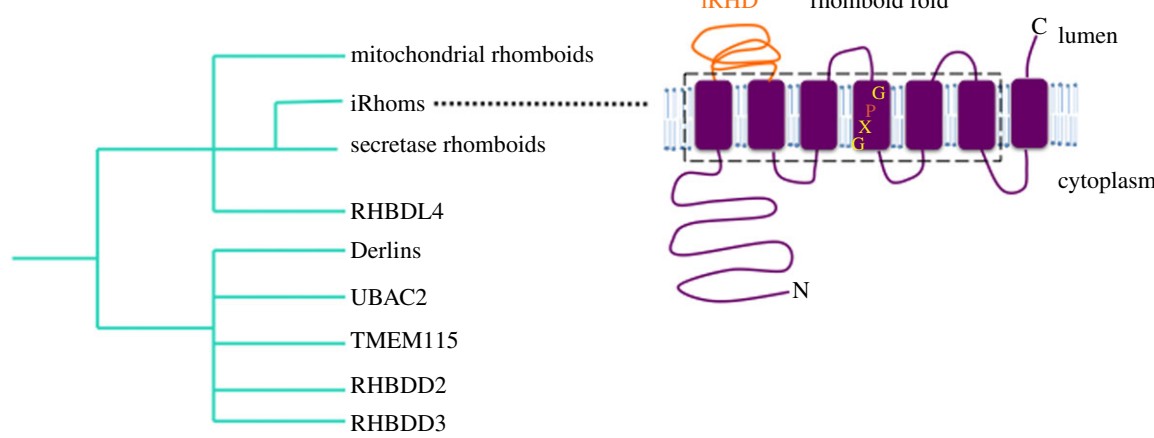

**Figure 1.** A family tree of the rhomboid-like superfamily. This schematic, based on existing sequence analysis and functional data, illustrates the relationship shared by iRhoms with the active proteases and the other inactive pseudoproteases of the rhomboid superfamily. Note this is not an evolutionary model. Inset depicts the seven transmembrane topology of iRhoms, comprising a conserved Rhomboid fold (boxed area), an extended cytoplasmic N-terminal tail and a luminal loop (iRHD).

## 2. iRhoms

iRhom pseudoproteases lack essential rhomboid catalytic residues, but are nevertheless quite closely phylogenetically related to ancestral rhomboid proteases [7,10]. They have seven transmembrane domains (TMDs) and are predominantly localized in the endoplasmic reticulum (ER). iRhoms are conserved in metazoans: there is one in *Drosophila*, while mammals harbour two of them, namely iRhom1 (gene name *Rhbdf1*) and iRhom2 (*Rhbdf2*). In the location of what would be the active site of protease rhomboids, iRhoms have a conserved proline immediately N-terminal to the expected location of the catalytic serine, (i.e. GPx replaces the rhomboid catalytic motif of GxS). They also possess a long N-terminal cytoplasmic domain and a highly conserved, luminal, cysteine-rich 'iRhom homology domain' (IRHD) linking TMD1 and TMD2 [7,10] (figure 1). In mammals, iRhom1 is expressed in many tissues, whereas iRhom2 expression is more limited, mostly to immune cells and skin [11,12]. iRhom2 knockout (KO) mice are fertile and viable [13,14]. The phenotype of iRhom1 KO mice is less clear because one study showed a severe phenotype with defects in several organs/tissues in three different strains tested [11], while another mouse model had a much weaker phenotype [12]. This disparity could potentially be due to differences in the genetic backgrounds used, or perhaps due to the possible presence of functional shorter forms of iRhom1 in the mutation which deleted only exons 4–11 [12], potentially making this not a complete null mutation. Conversely, the more severe phenotype, which was caused by deleting exons 2–18 deletion [11], might in principle affect sequences that control neighbouring genes, contributing to the observed phenotype. But in unpublished work from our group, we have found no evidence for the latter. Regardless of this uncertainty about the iRhom1 null phenotype, a rather bewildering number of cellular functions have now been reported for iRhoms, and in the next sections we will try to describe them and develop some common themes.

### 2.1. iRhoms and protein turnover

The first insight into the physiological function of iRhoms came from genetic studies in *Drosophila* [15]. The single fly

iRhom regulates epidermal growth factor receptor (EGFR) signalling by inducing the degradation of EGF-like ligands through a process resembling ER-associated degradation (ERAD), an important protein quality control mechanism (figure 2). The detailed mechanism how iRhoms interacts with protein degradation machinery is yet to be fully resolved. This study also showed that mammalian iRhoms can induce the proteasomal degradation of similar ligands, indicating that this function is potentially conserved. This degradation capability of iRhoms appeared specific for EGF-like proteins [15], but whether this proposed role of iRhoms affects only the EGFR signalling pathway and, if so, what determines this specificity, are yet to be determined. Moreover, any physiological relevance in mammals remains unknown, although there is a developing theme of iRhoms being involved in the regulation of protein stability and turnover. For example, iRhom1 has been reported as a regulator of proteasome activity under ER stress conditions in both human cells and flies [16]. Absence of iRhom1 prevents the dimerization of proteasome assembly chaperone 1 and 2 (PAC1 and PAC2), leading to impaired assembly and function of the 26S proteasome complex. Whether this proposed function of iRhom1 in regulating the turnover of cytoplasmic proteins (Huntingtin mutant and a GFP degron) [16] is related to its ability to degrade EGFR ligands at the ER is unclear. In another case, iRhom1 was reported to control the level of the transcription factor hypoxia-inducible factor-1α (HIF1α) via an oxygen-independent degradation process involving receptor of activated protein C kinase-1 (RACK1) [17]. RACK1 recruits E3 ubiquitin ligase complexes to promote HIF1α ubiquitination and degradation, and iRhom1 inhibits the interaction of RACK1 to HIF1α via competitive binding. But, developing a theme of much of this early discovery research, the physiological role of HIF1α regulation by iRhom1 remains unclear.

In addition to supporting protein degradation, iRhoms can also regulate protein turnover by stabilizing some client proteins. STING is a central adaptor in the innate immune response to DNA viruses [18]. Upon sensing viral DNA, STING traffics from the ER to the perinuclear microsomes, thereby activating IRF3 transcription pathways to induce expression of type I interferons [19]. In uninfected cells or the early phase of infection, iRhom2 acts as an adaptor

royalsocietypublishing.org/journal/rsob  Open Biol. **9**: 190003

**Figure 2.** The multi-faceted roles of iRhoms in protein turnover. An illustration of the role played by iRhoms in driving or protecting its clients from proteasomal degradation. The section on the left depicts EGF (blue) in the endoplasmic reticulum (ER) being driven towards the proteasome by *Drosophila* and mammalian iRhoms for its degradation. On the right, is an illustration of iRhom2 protecting STING from proteasomal degradation by recruiting the de-ubiquitinating enzyme, EIF3S5 (green) to the ER, in uninfected cells or early stages of DNA virus infection.

protein, promoting the interaction of STING and EIF3S5, a deubiquitinating enzyme, thereby inhibiting the degradation of STING (figure 2) [18]. This allows infected cells to elicit the appropriate immune response against the invading DNA virus. Similarly, iRhom2 is reported to regulate the stability of the mitochondrial membrane-located protein VISA, an essential adaptor protein in innate immune response to RNA viruses [20]. Upon virus infection, VISA regulates TLR3-triggered NF-κB and IRF-3 activation pathways [21]. In uninfected and early-infected cells, iRhom2 inhibits degradation of VISA by RNF5, an ER-localized E3 ubiquitin ligase by downregulating RNF5 level. In late phases of viral infection, iRhom2 interacts and promotes the degradation of MARCH5, a mitochondrial E3 ubiquitin ligase targeting VISA [20], although the mechanism by which predominantly ER-localized iRhom2 could interact with MARCH5 has not been established.

In summary, there is an emerging and quite convincing theme of iRhoms participating in the control of protein stability in multiple contexts. The proposed mechanisms, however, are diverse, and it is too early to conclude whether these examples represent a genuinely conserved function or are just disparate examples that might have evolved separately.

## 2.2. iRhoms and membrane protein trafficking

The first functional reports of iRhoms in mammals described the role of iRhom2 in controlling inflammation. Loss of iRhom2 inhibits the release of the primary inflammatory cytokine TNFα in response to stimulation with LPS [13,14]. This was shown to arise from failure of the membrane-tethered protease ADAM17—also called TNF alpha converting enzyme (TACE)—to migrate from the ER to the Golgi, where it is cleaved by furin protease and subsequently trafficked to the cell surface (figure 3). Consequently, iRhom2-

deficient mice are unable to elicit TNFα response, are more resistant to LPS-induced septic shock, and are also less efficient in controlling infection by *Listeria monocytogenes* [13,14]. iRhom2 was therefore proposed to act as a cargo receptor for ADAM17, responsible for its trafficking from the ER. As detailed in the next section, it has since become clear that the relationship between iRhom2 and ADAM17 is more long-lived, with iRhom2 accompanying and regulating ADAM17 throughout its lifetime. iRhom1 also regulates ADAM17 [11,12], and current evidence suggests that, in this context at least, iRhom1 and iRhom2 have similar functions, albeit acting in different cell types, reflecting their respective expression [11–14,22,23]. An essential point to emphasize here is that iRhom1 and iRhom2 are jointly responsible for all ADAM17 activity. This means that iRhoms underlie all ADAM17 functions, including not only its role in TNFα release, but also its role in the release of many of the EGFR ligands and, indeed, multiple other substrates. Although this review will focus substantially on inflammatory functions, this is a consequence of the historical development of the field and in the future, with further discovery of new client proteins, it may be that other iRhom roles prove to be as or even more pathophysiologically significant.

As described earlier, an example of an iRhom role not associated with inflammation (or indeed ADAM17 at all) is its modulation of the cellular immune response of cells to DNA viruses by regulating the turnover of the membrane protein STING [18]. Beyond its role in regulating STING stability, and in a process that appears somewhat similar to its role in ADAM17 trafficking, iRhom2 also facilitates the infection-triggered trafficking of STING, from the ER to perinuclear microsomes via the Golgi (figure 3) [18]. Moreover, iRhom2-mediated trafficking of STING was inhibited by human cytomegalovirus (HCMV) tegument protein UL82, thereby blocking the host innate immune response to DNA virus infection [24].

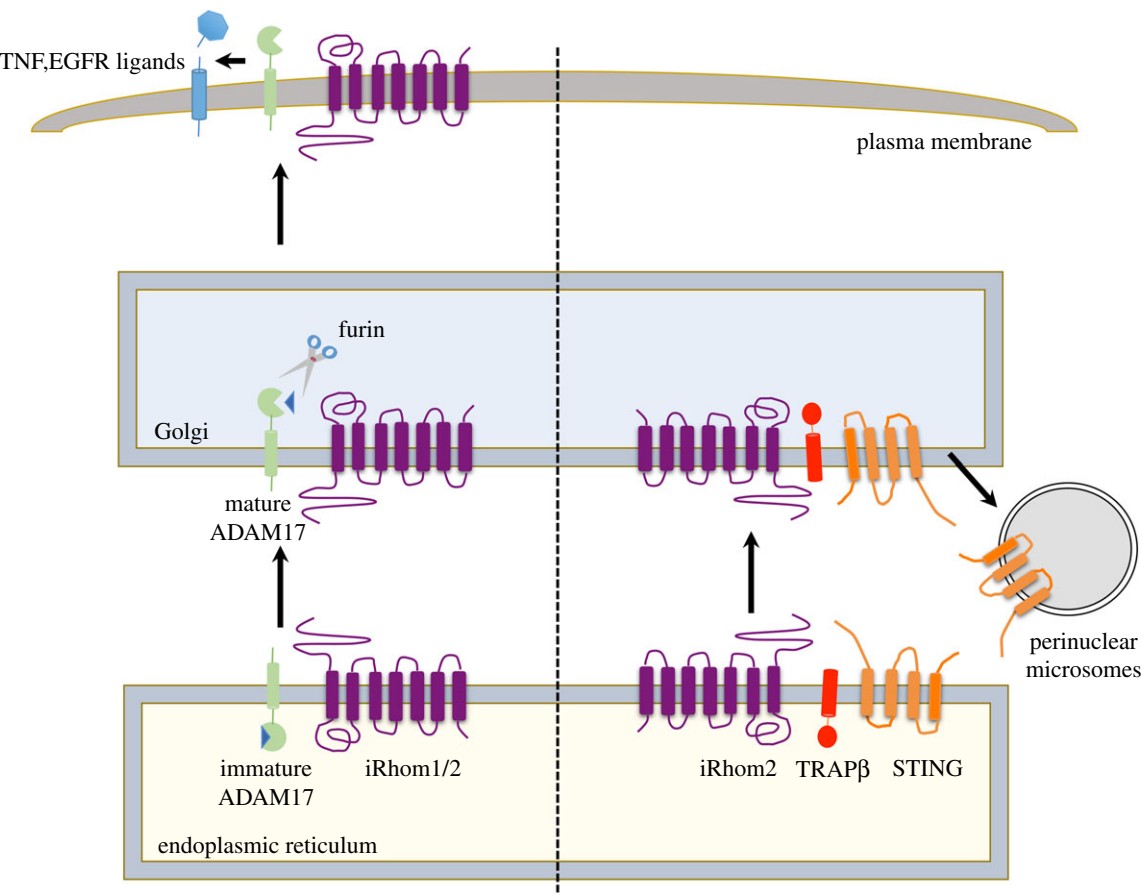

**Figure 3.** iRhoms in protein trafficking. A schematic of how iRhoms play an important role in the trafficking of the membrane proteins, ADAM17 (green) and STING (orange). On the left is an illustration of iRhom1 and 2 promoting the trafficking of ADAM17 from the ER to the Golgi apparatus, where the latter undergoes maturation by furin mediated cleavage to remove its prodomain. iRhoms further aid in the movement of ADAM17 from the Golgi to the plasma membrane, to promote shedding of TNF and EGFR ligands (blue). On the right is a depiction of iRhom2 facilitating the movement of STING from the ER to the microsomes via the Golgi, with the aid of TRAPβ (red).

These results show that iRhoms can regulate the cellular trafficking of at least two membrane proteins (ADAM17 and STING). It is not yet clear whether the trafficking of ADAM17 and STING by iRhoms share a similar mechanistic pathway, but this work raises the prospect that iRhoms may similarly control the trafficking of additional unknown and likely membrane client proteins. Furthermore, the data also show clearly that iRhoms have dual effects on both EGFR ligands and STING [13,15,18], via their distinct abilities to control the cellular trafficking and protein turnover of these clients. How the decision-making process of iRhoms is determined and regulated, for example whether client proteins are exported, degraded or stabilized, remains to be investigated.

# 3. Regulation and mechanistic insights into iRhom functions

## 3.1. Structure and function of iRhoms

A useful way to get a better understanding of the regulation of iRhoms is to break down the protein into different 'modules', namely the transmembrane, N-terminal cytoplasmic, and luminal domains. Below we discuss this modular regulation of iRhoms and how it contributes to their different cellular functions.

### 3.1.1. Transmembrane domains

The unifying characteristic of the rhomboid-like superfamily of proteins is a conserved 6-TMD core, also known as the rhomboid-like fold. Variation between clan members derives from some having a 7th TMD, the varying length of N- or C-terminal cytoplasmic domains, and the common existence of extended luminal/extracellular domains [7,25] (figure 1). This modular organization combined with what mechanistic and structural information we have so far from rhomboid proteases, leads to a view that the core function of the rhomboid-like fold is the specific recognition of TMDs [26]. In the case of bacterial rhomboid proteases, it is well established that substrate transmembrane helices are specifically recognized by rhomboids, and that this is the primary site of interaction [27]. Since all iRhom clients identified to date also have transmembrane domains, and considering the rather close evolutionary relationship between iRhoms and the rhomboid proteases, it seems likely that the iRhom–client interaction mimics the rhomboid protease–substrate interaction. Consistent with this, the mouse *iRhom2^sinecure* mutation, which blocks ADAM17 maturation and activation is a point mutation in the first TMD (I387F) (figure 4) [28,29]. Molecular dynamics simulation of the first iRhom2 TMD predicts that this I387F mutation is located in the middle of the transmembrane helix, which is slightly tilted in the membrane bilipid layer [28]. It is speculated that this could provide an interface for interaction with client TMDs.

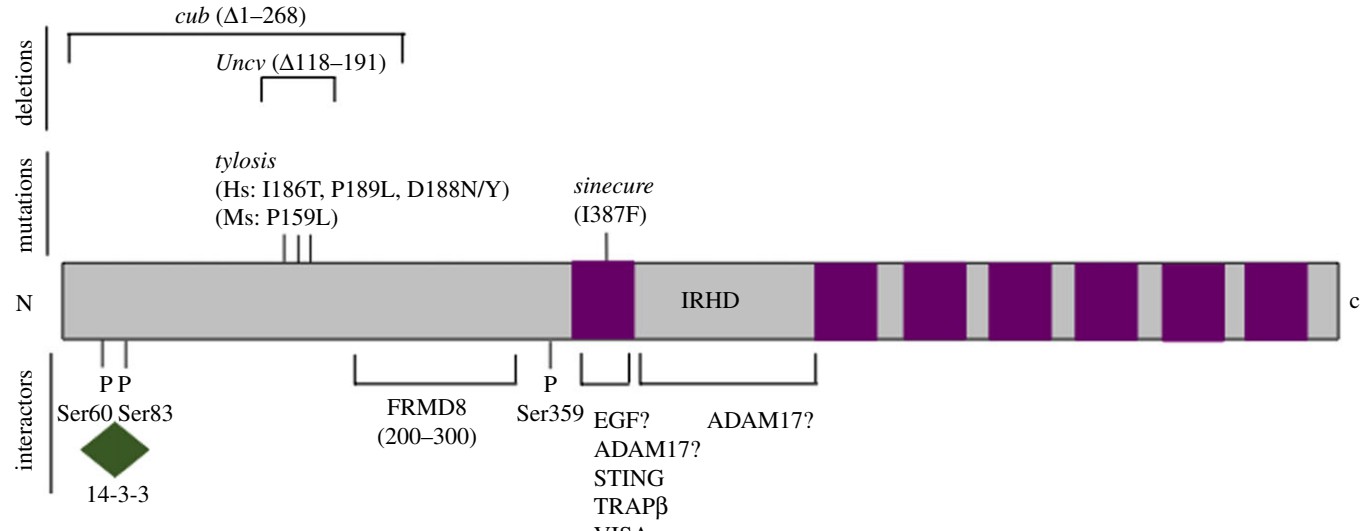

**Figure 4.** Domain architecture of iRhom2 with its interactors and phosphorylation sites. This figure depicts the transmembrane domains (purple) of iRhom2 along with a highly conserved iRhom Homology domain (IRHD) and the long N-terminal cytoplasmic tail, which harbours a number of functionally important 14-3-3 binding sites (in green). This schematic also highlights a number of known mapped mutations on the N-terminal, some of which are implicated in human disease, and proposed binding sites for known clients/interactors of iRhoms.

The observation that iRhom2 binding to VISA, a single-pass transmembrane protein, requires the first TMD of iRhom2 [20], strengthens the case for the significance of the first TMD. In the case of interaction with STING, a multi-pass transmembrane protein, the first TMD of iRhom has also been shown to be essential [18]. Overall, a clear emerging theme is the importance of the first TMD of iRhoms in several of their functions. Note, however, that there is also evidence for other domains being involved, at least in some cases. It was shown that the N-terminal of iRhom2 is required for binding to mature ADAM17 and the IRHD domain for interaction with the immature form of ADAM17 [23]. Moreover, the interaction of iRhom1 and TGFα is dependent on the latter's EGF-like ectodomain [30], implying that TMD–TMD interactions are not always the only determinants of iRhom–client interactions.

### 3.1.2. N-terminal cytoplasmic domain

One of the most conspicuous features of iRhoms is their extended N-terminal cytoplasmic domains, which have important but not yet well-defined regulatory functions. This domain has proven to be quite complex with both positive and negative regulatory elements for iRhom functions. Mutations in a highly conserved area of the iRhom2 cytoplasmic domain cause the rare inherited cancer syndrome Tylosis with oesophageal cancer (TOC) [31–33] (figure 4). They lead to enhanced activity of ADAM17 and constitutive shedding of EGFR ligands [34], although the mechanistic significance of the 4 amino acids region where all TOC mutations occur is not yet clear. The mouse *curly bare* (*cub*) mutation is a genomic deletion of most of the cytoplasmic domain of mouse iRhom2 [35,36]. Although there is some disagreement about its effect, evidence supports the idea that it has modestly elevated constitutive activity but is no longer inducible by LPS. Interestingly, TOC and *cub* mutations also lead to an increase in iRhom2 protein stability [35], which might contribute to their phenotypes. The idea that there is some inhibitory regulatory function within the iRhom2 cytoplasmic domain is further supported by the

observation that its deletion causes elevated constitutive ADAM17-dependent shedding of TNFR1 and TNFR2 [34].

But the iRhom2 cytoplasmic domain is not only a negative regulator of function: it is also required for full iRhom2 activity (table 1). The *cub* mutant and iRhom2 lacking the entire N-terminal domain can both still induce ADAM17 maturation, but in this case ADAM17 is quickly degraded via the lysosome, indicating the role of the cytoplasmic domain of iRhom2 in stabilizing mature ADAM17 [23,36]. It was also reported that iRhom2 lacking its cytoplasmic domain cannot rescue the defect in PMA-stimulated Kit Ligand 2 shedding in iRhom2−/− MEFs [37]. Intriguingly, no difference is observed for TGFα shedding, indicating a possible role of N-terminal of iRhom2 in determining substrate selectivity of ADAM17.

There is beginning to be some molecular insight into the regulatory functions of the cytoplasmic domain. At the plasma membrane, upon stimulation by GPCRs or PMA, the cytoplasmic domain of iRhom2 is phosphorylated on well-defined sites, leading to the recruitment of 14-3-3 proteins [23,40]. 14-3-3 recruitment is necessary and sufficient to trigger ADAM17-dependent shedding [23,40]. By contrast, phosphorylation-defective mutants still support constitutive shedding by ADAM17, indicating that phosphorylation controls specifically the stimulated shedding of ADAM17 substrates [23,40]. It is noteworthy that the cytoplasmic domain of ADAM17 is dispensable for its rapid stimulated activity [41,42] and the N-terminal domain of iRhoms instead appears to fulfil this regulatory role. Moreover, as described above, deletion of the iRhom cytoplasmic domain does not inhibit ER to Golgi trafficking of ADAM17, underscoring the separable regulatory functions.

Recently, a new binding partner of iRhom2 was identified, further illuminating the molecular details of its regulation. The poorly characterized FERM domain containing protein FRMD8 (also called iTAP) binds to a specific region of iRhom2 between amino acids 200 and 300 [22,43]. FRMD8 is important for the maturation of ADAM17 and its shedding activity at the cell surface and has been shown to stabilize both members of the iRhom2/ADAM17 complex at the cell surface [22,43]. It is noteworthy that iRhom2

**Table 1.** A list of mutations in iRhom2, with their corresponding functional effects on ADAM17 maturation and shedding processes.

| mutations in iRhom | effects on ADAM17 activation | | |
| --- | --- | --- | --- |
| | maturation | constitutive shedding | induced shedding |
| ΔN (N-terminus deletion) | reduced [23,34] | increased [34][a] <br> no difference [37] | reduced [34,37] |
| ΔiRHD (iRhom Homology domain deletion) | reduced [23] | n.a. | n.a. |
| cub | reduced [36] | increased [35][b] <br> reduced [36] | reduced [36] |
| Tylosis | increased [38] | increased[38] <br> increased [34][a] | n.a. |
| sinecure | reduced [28] | increased [28][c] | reduced [28,29] |
| uncovered | reduced [39] | reduced [39] | n.a. |

[a]Increased TNFR shedding.
[b]Greater levels of AREG secretion independent of TACE activity.
[c]KitL2 only.

binding to FRMD8 appears independent of 14-3-3 proteins, but nothing is yet known about any relationship between phosphorylation-dependent 14-3-3 binding and stabilization of the complex by FRMD8 [22,43].

Although it has been less studied, iRhom1 regulation shares several of the same characteristics as iRhom2. FRMD8 binds to both iRhoms, and the phosphorylation sites required for 14-3-3 binding are also conserved in both [22]. There are, however, no TOC-like disease mutations yet reported to affect iRhom1. This might be a consequence of the much wider expression of iRhom1, which is likely to make mutations lethal, but it is also notable that the cytoplasmic N-terminal is the least conserved domain between iRhom1 and iRhom2 (42% between human iRhom1 and iRhom2), raising the possibility that there may be distinct regulatory functions.

Overall, current evidence highlights the regulatory importance and complexity of the N-terminal cytoplasmic domains of iRhoms, and the existence of distinct sub-regions within this domain. On a more specific note, we now have a picture of iRhom2 (and probably iRhom1) having an intimate and long-lived relationship with ADAM17. Whereas the initial reports suggested that iRhom2 was a cargo receptor of some kind, which supported the trafficking of ADAM17 from the ER to the Golgi [13,14], we now know that later in ADAM17's lifetime, iRhom controls its stimulated release of substrates from the cell surface, as well as the stability of the iRhom2/ADAM17 complex [22,23,40,43]. It was also shown that iRhom2 can in some way control the substrate specificity of ADAM17, influencing the cleavage of some but not all of its substrates [37]. Based on the hypothesis that the structural relationship between iRhoms and ADAM17 is similar to that of rhomboid proteases and their substrates, it has been proposed that iRhoms interact with ADAM17 via TMD1, and that TMD2-TMD5 create the interface for a transient and selective interaction with ADAM17 substrates [26]. More structural studies are needed to validate this model, but it is certainly helpful to think of iRhom2, supported by FRMD8, as being regulatory cofactors, or subunits, of an ADAM17 sheddase complex.

### 3.1.3. iRhom homology domain

The extended luminal/extracellular loop domain between TMDs 1 and 2, known as the iRhom homology domain (IRHD) is intriguing and rather mysterious. It is the most highly conserved domain of iRhoms (65% between human iRhom1 and iRhom2), but its function is still unknown [9,10]. The IRHD is about 230 amino acids long and contains 16 conserved cysteine residues that are predicted to exist as 8 disulphide bonds, implying a complex three-dimensional structure. No structure for the IRHD (or indeed any part of an iRhom) has yet been solved and it is not predicted to resemble any other known domain. Deletion of the iRhom2 IRHD prevents ADAM17 maturation [23], and this is attributed to decreased binding to immature ADAM17 in the ER (figure 4). This suggests that the TMD interaction between iRhom2 and ADAM17 (at least for the immature form) may not be the only interface, and that the IRHD mediates a luminal interaction. iRhom2 with the IRHD deleted is detected at similar levels at the cell surface to wild-type [23], indicating that the IRHD is likely to be dispensable for iRhom2's own trafficking.

## 3.2. Post-translational modifications

We have discussed the role of phosphorylation in the stimulated shedding of ADAM17 substrates [23,40]. An earlier study also showed that nitric oxide synthase (iNOS) expression regulates the shedding of TNF receptor by ADAM17 upon LPS exposure in hepatocytes [44]. The trafficking and activation of ADAM17 to the plasma membrane was associated with serine phosphorylation of iRhom2 (and ADAM17), in this case reported to be dependent on nitric oxide-induced protein kinase G. These examples underscore the potential diversity of phosphorylation-dependent regulation of iRhoms. Beyond the 14-3-3 proteins that mediate some of the reported response to phosphorylation, mass spectrometry analysis shows a number of other phosphorylation-dependent proteins bound to

royalsocietypublishing.org/journal/rsob    Open Biol. 9: 190003

iRhom2 [40], suggesting the existence of further unknown phosphorylation-dependent functions of iRhoms.

iRhoms have also been shown to be regulated by ubiquitination, another important regulatory modification. The half-life of iRhom2 is extended in the presence of the proteasomal inhibitor MG-132 [35], which was not the case for the *cub* mutant of iRhom2, suggesting that the cytoplasmic domain may contain sites for K48-ubiquitination, and therefore proteasomal degradation. One of the critical components of TNFα-induced NF-κB signalling pathway is the E2 complex Uev1A-Ubc13 [45]. It mediates K63-linked ubiquitination of RIP1 that acts as a scaffold to recruit NEMO (NF-κB essential modulator) to activate the downstream pathway. It was recently shown that in response to TNFα, Uev1A-Ubc13 with the cooperation of CHIP E3 ligase, adds K63-linked polyubiquitin chains to iRhom2 [46]. This leads to increased ADAM17 maturation and cleavage of the TNFR, thereby acting as a negative feedback mechanism to inhibit TNFα-induced NF-κB signalling. As iRhom2 associates with ADAM17 throughout the secretory pathway and regulates both the maturation and shedding processes, it is not clear at which point K63-polyubiquitinated iRhom2 might be having an effect. Moreover, how K63 polyubiquitination on iRhom2 enhances its activity, and where this modification occurs on iRhom2 is not known. It is noteworthy that CHIP E3 ligase is also a regulator of proteostasis [47], but whether K63 polyubiquitination has a role in iRhom-mediated degradation of EGFR ligands, or indeed in FRMD8-mediated stabilization of the iRhom2/ADAM17 complex at the plasma membrane [22,43] also remains to be investigated.

In summary, these post-translational modifications have important regulatory effects on the functions of iRhoms and it seems certain that their further investigation will yield a better understanding of iRhom biology.

## 3.3. iRhom expression

### 3.3.1. Transcriptional regulation

To date, most studies on the function and regulation of iRhom proteins have dealt with how they affect different signalling pathways. There has been relatively little investigation of the regulation of iRhoms themselves at the gene and transcriptional level. According to the UCSC genome browser, both iRhom1 and iRhom2 have many transcripts, with several of them able to code for alternative forms of the proteins. Due to the lack of reliable antibodies that can detect endogenous iRhoms, the functional significance of this has not been much examined. It was recently shown, using RNA sequencing, that two isoforms of iRhom2 are expressed in cancer-associated fibroblasts, with isoform 2 predominating in cancer cells [48]. Isoform 2 was also reported to be more potent than isoform 1 at activating the TGFβ signalling pathway and promoting cell motility, suggesting that different isoforms might have distinct functions. These are the only two annotated isoforms on the NCBI database, and they differ by the loss of amino acids 50–79 in the N-terminal cytoplasmic domain in isoform 2. In these studies, the activation of TGFβ signalling is proposed to depend on ADAM17 trafficking and subsequent cleavage of TGFβR1 [48], so this work suggests the possible presence of an ADAM17 regulatory site between amino acids 50–79 of iRhom2. This is interestingly reminiscent of

the increased ADAM17 activity observed with the *cub* and TOC iRhom2 mutants discussed earlier.

Beyond the apparent existence of distinct isoforms of iRhoms, several studies have shown that iRhom transcription can be regulated. Exposure to lipopolysaccharides (LPS), which induces ADAM17 activity by iRhoms, also leads to a rapid transcriptional upregulation of iRhom2 in macrophages [13,14]. Similarly, in addition to its role in STING trafficking and degradation in response to DNA virus infection, iRhom2 transcript level is robustly increased within hours of infection in human monocytic THP-1 cells and murine bone-marrow-derived macrophages (BMDMs) [18]. According to ENCODE transcription factor ChIP-seq data (illustrated by the UCSC Genome Browser), there are two binding sites in the promoter region of iRhom2 for NF-κB, a transcription factor activated by both LPS and viral infection. ER stress inducers such as tunicamycin and thapsigargin increase iRhom1 mRNA level within hours of exposure, leading to its function in proteasome assembly and degradation [16].

The importance of further investigation of the transcriptional regulation of iRhoms is reinforced by the observation that iRhom2 is among the top genes to be methylated in Alzheimer's disease [49,50]. This type of modification is typically associated with transcriptional silencing of the genes. Furthermore, upregulation of iRhom1 transcripts has been observed in breast cancer tissues [51]. It is clear that, although not yet well investigated, transcriptional regulation of iRhoms represents an important layer of their regulation physiologically and in disease.

### 3.3.2. Cell and tissue expression

Most iRhom expression data is at the level of RNA transcripts. *Drosophila* iRhom is only detectable in the central nervous system of embryos, particularly in the ventral nerve cord and brain [15], and its expression remains relatively restricted to neural tissues such as optic lobes, retina and brain throughout development. In mammals, iRhom1 is expressed in most tissues, whereas iRhom2 expression is much more limited, particularly to myeloid cells [11,13,14]. This differential distribution explains the phenotypic differences between iRhom1 and iRhom2 knockout mice, where the former has a more severe phenotype [11]. As well as bone-marrow-derived macrophages, iRhom2 expression is also high in microglia (brain-resident macrophages) [12,52] and Kupffer cells (liver-resident macrophages) [53]. In all these cases, iRhom2 regulates ADAM17 activity. It is noteworthy that in the brain there is a clear separation of expression, with iRhom1 present in all cell types except for microglia, which express high levels of iRhom2 [52].

Although early reports emphasized the relative specificity of iRhom2 expression in myeloid cells, it later became clear that there are functionally important levels in some other tissues as well including, for example, skin and lungs [22,32]. Interestingly, some of these tissues also have high iRhom1 mRNA levels, and this highlights an important general point. Although there are some cell types like macrophages and certain brain cells, in which either one or the other iRhom is specifically expressed, many cells express both. As referred to above, this leads to important unanswered questions about whether iRhom1 and iRhom2 have identical or redundant functions, implying that the differences in their biology is simply due to their different expression patterns,

or whether there are significant mechanistic distinctions between the two proteins.

# 4. iRhoms and disease relevance

One important consequence of the discovery that iRhoms are critical regulators of ADAM17 is their role in many inflammatory diseases, primarily mediated by TNFα signalling. However, many other pathological conditions are emerging as potentially linked to iRhom biology, including cancer, infection, neurodegeneration, and skin and heart diseases. In a few cases there is direct evidence of iRhom association with disease; in others the evidence is more indirect, stemming from the cellular models. A summary of potential disease association is shown in table 2.

## 4.1. iRhoms and cancer

The EGFR and TNFα signalling pathways are highly implicated in tumour growth and development [62], and considering the integral function of iRhoms in these pathways, one might expect a high frequency of deregulation of iRhoms. Indeed, one of the earliest papers mentioning iRhoms reported that iRhom1 is essential for the growth of some epithelial cancer cells [51]. iRhom1 is highly expressed in early-stage breast cancer and its deletion leads to apoptosis and autophagy of breast cancer cell lines, as well as a reduction in xenograft tumour growth [51]. This potential role of iRhom1 is further supported by the significant correlation between its elevated expression and different clinical measures of breast cancer namely metastasis, poor response to chemotherapy and decreased survival [17]. Another link of iRhom1 to cancer is illustrated by its role in regulating GPCR-ligand-induced growth, proliferation and invasion of head and neck squamous cancer cell lines and xenograft tumours [63]. In this work, knockdown of iRhom1 led to reduced EGFR activation, and it was proposed that iRhom1 is the link between GPCR activation and EGFR signalling. The potential of iRhom1 as an oncogene was further underlined by a report that iRhom1 level is significantly upregulated in colorectal cancer [54]. In this case, it was proposed that iRhom1 influences components of the Wnt/β-catenin signalling pathway to promote epithelial-to-mesenchymal transition (EMT) and cell proliferation.

We have already described that mutations in the iRhom2 gene cause the rare autosomal dominant disease Tylosis with oesophageal cancer (TOC). This disease is typified by palmoplantar hyperkeratosis and a high risk of developing oesophageal cancer in middle age. The first mutations were found in UK and US (Ile186Thr) and German (Pro189Leu) families [32]. Subsequently, more mutations in iRhom2 associated with TOC were found in Finnish (Asp188Asn) [31] and African (Asp188Tyr) [33] families. These are proposed to be gain-of-function mutants with increased EGFR ligand shedding, proliferation and migration potential observed in tylotic keratinocytes [32,34]. Additionally, the *cub* mutation of iRhom2, increases the susceptibility to adenoma formation and decreases survival in $Apc^{Min/+}$ mice, a model of human familial adenomatous polyposis [35].

Cancer-associated fibroblasts (CAFs) are a major component of the tumour environment and are important in supporting tumour growth and development in many cancers [64]. iRhom2 is highly expressed in CAFs and is reported to regulate ADAM17-dependent cleavage of the TGFβ receptor TGFBR1, contributing to the progression of diffuse-type gastric cancers (DGCs) [48].

Overall, the evidence is strong that both iRhoms are implicated in a wide range of cancers. The only case where there is a well-defined and direct disease mechanism is TOC, but in many more common and sporadic cancers, there is a growing body of work that supports an association. Given the role of iRhoms in controlling inflammatory and growth factor signalling, this involvement is not a surprise, but note that in most cases the mechanistic link between the iRhom and the pathology is not fully explored.

## 4.2. iRhoms and skin disease

As indicated earlier, hyperkeratosis precedes oesophageal cancer in TOC patients. Further insights into the role of these mutants were obtained by the generation of a mouse model for the human *iRhom2* Pro189Leu TOC mutation [55]. These mice have complete hair loss at birth but develop a thin curly hair coat as adults. The mouse skin also shows signs of hyperkeratosis and an abnormal wound healing phenotype, associated with increased secretion of amphiregulin (AREG) [55]. Genetic deletion of AREG in these mutant mice restores a normal skin phenotype, indicating that iRhom2-regulated AREG production is an important contributory factor to the skin phenotype of TOC. This work is closely paralleled by investigations of the mouse *cub* allele of iRhom2, which also has a hairless phenotype [65], and is also genetically modified by loss of amphiregulin. Similar phenotypic and genetic observations were reported in another study [35], albeit with differing molecular interpretation, highlighting the complex interplay between iRhom2 and AREG/EGFR signalling. In another parallel between TOC in humans and the mouse *cub* mutation, homozygous *cub* mice have an enhanced wounding response in an ear notch closure assay [66], with the NRF2-mediated oxidative stress, integrin receptor aggregation and the FcγR-mediated phagocytosis pathways upregulated in the *cub* mutant.

Another mouse mutation in iRhom2, *uncovered* (Uncv), is an internal deletion of the cytoplasmic N-terminal. It is smaller than but overlaps with the *cub* deletion, and also leads to a hair-loss phenotype [39]. The hairless phenotype of *Uncv* mice is due to abnormal hair shaft and inner root sheath differentiation [67]. These defects were attributed to the inability of iRhom2 mutants to mature ADAM17 and it was speculated that this might have downstream effects on the activation of Notch and Wnt signalling pathways. iRhom2 *Uncv* mice have reduced hair matrix proliferation, but exhibit hyperproliferation and hyperkeratosis in the interfollicular epidermis, along with hypertrophy in the sebaceous glands [68]. Interestingly, the hair follicle stem cells appear normal, suggesting that the iRhom2 mutant might regulate hair follicle differentiation.

Keratins are cytoskeletal proteins vital for providing the physical strength to skin. Using mouse models, Maruthappu and colleagues showed that iRhom2 can regulate the thickness of the epidermis of the footpad [69], and that this is related to its interaction with Keratin 16 (K16). Additionally, keratinocytes from patients with TOC show an upregulation of K16 levels, alluding to its potential contribution to palmoplantar thickening. It was also described that iRhom2 can be regulated by p63, a transcription factor implicated in

**Table 2.** The disease conditions and the physiological effects of loss of function of either iRhom1, iRhom2 in mouse and human tissues, together with the relevant clients.

| disease | gene | phenotypic readout | client |
|---|---|---|---|
| breast cancer | iRhom1 | metastasis, poor response to chemotherapy, reduced survival [17] | reduced EGFR activation |
| colorectal cancer | iRhom1 | reduced cell proliferation, migration and invasiveness of tumor [54] | components of Wnt-β catenin signalling |
| tylosis with oesophageal cancer (TOC) | iRhom2 | (i) palmoplantar hyperkeratosis, increased risk of oesophageal cancer<br>(ii) adenoma formation and decreased survival<br>(iii) complete hair loss of mice at birth<br>(iv) increased wound healing [31–33,35,55] | EGFR ligands |
| gastric cancer-associated fibroblasts | iRhom2 | diffuse type gastric ulcers [48] | TGF-β1,ADAM17 |
| inflammatory arthritis | | less joint swelling, lowered synovial inflammation, cartilage erosion [56] | n.a. |
| renal dysfunction | iRhom2 | significant protection against tissue inflammation, kidney damage [57] | reduced ADAM17, EGFR |
| haemophilic arthropathy (HA) | iRhom2 | reduction in osteopaenia, synovial inflammation [58] | n.a. |
| hepatic steatosis | iRhom2 | reduced inflammatory cytokines [53] | ADAM17, TNF-α |
| acute lung injury after intestinal ischaemia-reperfusion | iRhom2 | reduction in apoptosis [59] | ADAM17, TNF-α |
| Listeria monocytogenes infection | iRhom2 | (i) increase in granulomas in liver<br>(ii) rapid death post infection [14] | n.a. |
| HSV-1 infection | iRhom2 | defective innate immune response to DNA virus [18] | STING |
| RNA virus infections (Sendia, VSV) | iRhom2 | (i) quicker mortality<br>(ii) increased immune cell infiltration and damage to lungs [20] | VISA |
| heart diseases | iRhom1 & iRhom2 | (i) cardiac infarction<br>(ii) formation of thrombosis [60,61] | n.a. |
| neurological disease | iRhom2 | Alzheimer's (speculated) [49,50] | n.a. |

epithelial development [70]. TOC keratinocytes are resistant to p63-induced cell death induced by ultraviolet B rays, indicating a complex relationship between iRhom2 and p63 in keratinocyte biology and their response to stress.

## 4.3. iRhoms and inflammatory diseases

As is apparent from much of the discussion above, the tight regulation of ADAM17 and its substrates, particularly TNFα, by iRhoms is an important contributory factor to the pathologies of inflammatory diseases. iRhom2 expression is upregulated in synovial macrophages from rheumatoid arthritis (RA) patients compared to healthy controls [56]. Using K/BxN mouse RA model, this study shows that iRhom2 knockout mice are significantly protected from inflammatory arthritis as shown by less joint swelling, synovial inflammation and cartilage erosion. Furthermore, using a model of TNFα-mediated septic shock and liver damage, iRhom2 knockout mice show reduced TNFα secretion, more severe damage to liver architecture, and resistance to LPS lethality compared to wild-type mice [14].

Renal dysfunction is often a consequence of the autoimmune disease systemic lupus erythematosus (SLE), and ADAM17 substrates TNFα and heparin-binding EGF (HB-EGF) have been implicated as important mediators of this condition called lupus nephritis (LN). It was shown that deletion of iRhom2 in a mouse model for LN leads to reduced TNFα and EGFR signalling, and a significant protection against tissue inflammation and kidney damage compared to wild-type mice [57]. Additionally, iRhom2 and HB-EGF expression are increased in kidneys of both the mouse model and LN patients, supporting a role of iRhom2 and ADAM17 in LN pathology.

One of the major outcomes of haemophilia A is haemophilic arthropathy (HA), a degenerative joint disease characterized by TNFα-dependent inflammation of the joint and surrounding tissues as well as bone loss [71]. iRhom2 loss in a mouse model for HA led to a marked reduction in osteopaenia and synovial inflammation [58]. Moreover, bleeding in the joints, hypothesized to be causal to the observed inflammation, is also absent in iRhom2 deficient animals.

Neutrophils and macrophages are important players in both acute and chronic inflammatory diseases. In particular,

macrophage proliferation that is dependent on ADAM17-mediated shedding of macrophage colony-stimulating factor-1 (CSF-1) from both neutrophils and macrophages is a key contributory process in these inflammation states [72]. Consistent with what we know about iRhom2 function, it was shown, using both acute thioglycollate-induced peritonitis and high-fat induced chronic atherosclerosis models, that iRhom2 is critical for driving macrophage proliferation [73].

Exposure of wild-type mice to traffic-related airborne particulate matter (PM2.5) leads to hepatic steatosis, metabolic syndrome and dyslipidaemia, and this correlates with an increased expression of iRhom2, and elevated TNFα [53]. These effects are significantly reduced in iRhom2 knockout mice. Knockdown of iRhom2 in Kupffer cells (liver-resident macrophages) leads to decrease in inflammatory cytokines, suggesting a role for iRhom2/ADAM17/TNF-α in regulating hepatic inflammation and lipid metabolism in response to PM2.5 [53].

Acute systemic lung inflammation is often a serious complication of intestinal injury, and is associated with high levels of TNFα. Lung inflammation accompanied with a reduction in neutrophil activity, apoptosis and TNF-α levels are significantly reduced in iRhom2 knockout mice compared to wild-type mice after intestinal ischaemia-reperfusion [59].

In summary, the major role of iRhom2 in regulating ADAM17 activity and inflammatory signalling by TNF makes it a significant player in many inflammatory conditions. Indeed, it is expected to be involved in all TNF-mediated events, so there are likely to be many further reports of iRhom2 in human disease. For this reason, we expect there to be a growing pharmaceutical interest in the possibility of targeting iRhoms as potential targets in the huge anti-inflammatory industry.

## 4.4. iRhoms and infectious diseases

The central role of iRhoms in inflammatory signalling, combined with their known functions in the cellular responses to both DNA and RNA viruses, gives them a prominent role in controlling infection by multiple pathogens. The loss of TNFα secretion in response to lipopolysaccharide (LPS) in iRhom2 knockout mice makes them resistant to the toxic shock caused by a normally lethal LPS dose [13,14]. Moreover, since TNFα is crucial for defence against bacterial infections, iRhom2 loss leads to sensitivity to Listeria monocytogenes infection [14]. The increase in granulomas in the liver as well as higher bacterial titres in spleen, liver, kidney and brain several days post-infection highlights the vital role of iRhom2 in pathogen defence.

Conversely, the level of transglutaminase 1 (TGM1), an enzyme involved in epidermal barrier formation, is increased in keratinocytes of TOC patients, suggesting they potentially have increased epidermal barrier function [38]. As this barrier is critical for regulating bacterial infection, it was shown that indeed, keratinocytes from TOC patients were more resistant to Staphylococcus aureus infection compared with normal keratinocytes [38].

The iRhom2 client STING participates in the cellular immune response to DNA viruses [18]. iRhom2 regulates the level of STING throughout the infection process, and iRhom2-knockout mice have increased susceptibility to lethal doses of herpes simplex virus type 1 (HSV-1). Similarly, the iRhom2 client VISA is a crucial player in the immune response to RNA viruses [20]. Fibroblast and immune cells deficient for iRhom2 have a reduced induction of antiviral genes in response to several RNA viruses, including Sendai virus and vesicular stomatitis virus (VSV). iRhom2 knockout mice show a higher penetrance of symptoms and quicker mortality upon infection with VSV compared to wild-type mice [20]. Additionally, these infected mice have increased VSV titres in the liver and spleen coupled with immune cell infiltration and damage to the lungs.

## 4.5. Other pathologies

A theme that has run throughout this review has been that the central role of iRhoms in inflammatory and growth factor signalling associates them with a very wide range of physiological and pathological processes. Indeed, the way we have separated into sections the different pathologies is a bit artificial, since most that we currently know about are rooted in those underlying inflammatory and signalling mechanisms. Since research into iRhoms is a fairly young field, we predict that many other iRhom-related disease processes will emerge, but it is less clear whether these will include examples that relate to currently unknown molecular roles of iRhoms, or whether they will all fit under the umbrella of inflammation and growth factor signalling.

In the brain, the role of iRhoms may be related to inflammation. As described above, iRhom1 predominates in neurons and iRhom2 in microglia, which are related to macrophages. But it is also possible that in these cells iRhoms have other roles, for example associated with their cellular function in protein stability control. In Drosophila, loss of iRhom causes a neurological phenotype described as increased sleep-like behaviour, due to elevated EGFR signalling. There are no obvious neurological defects detected in iRhom1 and iRhom2 knockout mice [11,13], although since iRhom1 deficient mice die within weeks of birth (at least in one report), probably because of a highly penetrant brain haemorrhage phenotype [11], this may not be very informative.

The recent reports that iRhom2 is one of the top genes with differential level of CpG DNA methylation in Alzheimer's disease (AD), particularly in the early stages, provides an interesting mechanistic line to pursue [49,50]. Expression of iRhom2 is altered in AD and it is postulated that this might be related to its potential function in microglia and infiltrating macrophages, but there is currently no evidence to support this, nor any other possible explanation for the association with AD. Intriguingly, overexpression of both Drosophila iRhom and human iRhom1 in the fly model for Huntington disease reduces the characteristic rough-eye phenotype [16], but whether this has any mechanistic relationship to a potential role of iRhoms in neurodegenerative disease is unknown.

Two recent reports have associated iRhoms with cardiac pathology [60,61]. In both cases this appears to be related to inflammatory function. Exposure to LPS induces myocardial infarction, associated with a heightened inflammatory response. It has been reported that iRhom2 is a positive regulator of LPS-triggered inflammation in the cardiac muscles [61]. Silencing of iRhom2 leads to a significant reduction in the release of pro-inflammatory cytokines, and the dampening of the Toll-like receptor-4/Nuclear Factor kappa-B (TLR-4/NF-κB) signalling pathway [61]. In another example, the

**Figure 5.** Mass-spectrometry based identification of protein interactors for iRhom2. Proteins reported to interact with iRhom2 in two separate papers (by Cavadas *et al.* [40] and Künzel *et al.* [22]). The Venn diagram depicts a region of overlap that signifies a common set of clients for iRhom2, identified in both studies. The sub-cellular localization of the clients (based on UNIPROT prediction) is displayed alongside (see key).

death of cardiomyocytes after myocardial infarction leads to an acute inflammatory response to prevent irreversible damage to the heart cells [61]. Different populations of macrophages mediate both the inflammatory and ensuing reparative phases. Using *iRhom2* knockout mice, it was reported that, following myocardial infarction, iRhom2 is required for cytokine release from macrophages during both phases [60]. *iRhom2* knockout mice have reduced cardiac function and collagen deposition at the infarcted site, leading to reduced tissue repair potential and to higher mortality.

## 5. Conclusion and perspectives

Since their recognition as members of the rhomboid-like superfamily of proteins, iRhom pseudoproteases have become increasingly prominent. In evolving from their active counterparts, iRhoms have acquired new and important regulatory domains (cytoplasmic N-terminal and luminal IRHD). These features, in conjunction with the transmembrane domains, appear to act as modules which are integral to several of the ascribed functions of iRhoms. These modules are modified (phosphorylation, ubiquitination), act as docking sites for regulatory proteins (keratin, 14-3-3, FRMD8) and interface with client proteins (ADAM17, STING, VISA, EGFR ligands) to work together in a concerted manner to allow iRhoms to exert their full functions. It is noteworthy that the least understood module of iRhoms is the mysterious cysteine-rich IRHD,

forming the first luminal loop. Given that this is the most highly sequence-conserved part of iRhoms, unravelling its role is paramount to a better understanding of iRhoms.

The most clearly defined role of iRhoms is currently the regulation of ADAM17, which in turn is fundamental to the activity of TNF and other cytokines, thus propelling iRhoms to the centre-stage of inflammatory signalling. As inflammation underlies so may diseases, the appreciation of the pathological role of iRhoms is likely to increase fast. Indeed, this review has documented the rapidly growing association of iRhoms with many other pathologies; addressing the exact role of iRhoms in these diseases is critical for exploring any therapeutic potential of targeting iRhoms.

Interaction of iRhoms with their client proteins via the cytoplasmic N-terminus, transmembrane domains and luminal/extracellular domain provides a variety of interfaces that could be amenable to manipulation by pharmacological inhibitors. One such potential focal point is the binding of iRhoms and ADAM17. However, what is still critically lacking is a detailed structural understanding of this interaction, so solving the structure of iRhoms in complex with ADAM17 is a current priority. Since iRhoms have evolved from their protease ancestors [5,10], one can speculate that iRhoms might interact with their client proteins in a manner similar to the rhomboid proteases and their substrates. However, unlike active proteases with catalytic pockets which can be targeted relatively easily, it may be more challenging to inhibit a potentially broader interaction interface between the TMDs of ADAM17 and iRhoms.

A few of the major evolving concepts of iRhom biology that remain to be resolved are: (i) the mechanistic basis for the clear involvement of iRhoms in regulating protein turn-over of membrane proteins (EGFR ligands, STING, VISA) and potentially cytoplasmic proteins (Huntingtin and HIF1α); (ii) the relationship between the dual functions of iRhoms in both cellular trafficking and degradation; and (iii) the range of different clients with which iRhoms work: the growing list of new interactors/clients (figure 5), points towards the existence of yet-to-be-defined functions of iRhoms in regulating other intracellular signalling pathways.

Data accessibility. This article has no additional data.
Competing Interests. We declare we have no competing interests.
Funding. Work in the Freeman group is supported by the Wellcome Trust (101035/Z/13/Z). S.M. has been supported by a Long-term fellowship (LTF) from HFSP and a Non-Stipendiary fellowship from EMBO.
Acknowledgements. We thank Adam Grieve and Shaked Ashkenazi for their advice about the preparation of this review.

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
