## [Reviewer comments · Open Biology]

Review History

RSOB-19-0003.R0 (Original submission)

Review form: Reviewer 1

Recommendation

Accept with minor revision (please list in comments)

Are each of the following suitable for general readers?

- a) **Title**
Yes
- b) **Summary**
Yes
- c) **Introduction**
Yes

Is the length of the paper justified?

Yes

Should the paper be seen by a specialist statistical reviewer?

No

Is it clear how to make all supporting data available?

Not Applicable

Is the supplementary material necessary; and if so is it adequate and clear?

Not Applicable

Do you have any ethical concerns with this paper?

No

Comments to the Author

The review article by Dulloo et al reviews the current knowledge of the molecular, cellular and pathophysiological role of iRhom pseudoproteases. The group of Prof. Freeman has been instrumental in uncovering the various activities of iRhom proteins and the authors did a magnificent job in summarizing these findings of the past 20 years. iRhoms are crucially needed for the trafficking and activation of the important ADAM17 protease, which governs at least three important pathways, namely the TNF α , IL-6 and EGF-R signaling pathways. These pathways are involved in the regulation of inflammation, infection, metabolism and cancer. In addition, there are additional clients of iRhoms, which are not very well studied yet, and which will possible change the direction of the field considerably. All important aspects of iRhom biology are nicely covered in the review.

The review article is very well written and it will appear timely since more and more aspects of inflammation and cancer research seem to involve iRhom activities.

There are some points, the authors might want to consider.

1. In the abstract, the authors write that the function of ADAM17 is shedding of cytokines and growth factors. ADAM17 is also an important sheddase of many receptors and adhesion proteins. This could be mentioned here.
2. On p3, the authors describe the different phenotypes of iRhom1 knock-out mice. Do the authors have a possible explanation for the observed variation on phenotypes?
3. On p5, the authors describe the role on iRhom2 in the trafficking of ADAM17 from the ER to the Golgi, where ADAM17 is processed by furin proteases. In contrast to the statement of the authors, this furin cleavage does not lead to activation of ADAM17. This statement should be rephrased.
4. On p12, the authors describe the transcriptional activation of iRhoms. They might want to include a short statement on the structure of the promoters of the iRhom genes, which might illustrate which pathways are likely to affect transcription of iRhom genes.
5. Some references (20, 21, 33, 39, 65, 67, 70) are incomplete and should be amended.
6. In the inset of Fig. 1, the authors might want to indicate the iRhom homology domain (IRHD).
7. As far as this reviewer understands, the data summarized in Fig. 5 are not from Cavadas et al but rather from Oikonomidi et al. This should be corrected.

Review form: Reviewer 2

Recommendation

Accept with minor revision (please list in comments)

Are each of the following suitable for general readers?

- a) **Title**
Yes
- b) **Summary**
Yes
- c) **Introduction**
Yes

Is the length of the paper justified?

Yes

Should the paper be seen by a specialist statistical reviewer?

No

Is it clear how to make all supporting data available?

Not Applicable

Is the supplementary material necessary; and if so is it adequate and clear?

Not Applicable

Do you have any ethical concerns with this paper?

No

Comments to the Author

The iRhom pseudoproteases are emerging as important regulators of the metalloprotease ADAM17, which has a critical role in the body's first line of defence against infection (via shedding of TNF α) and repair (via shedding of EGF receptor ligands). This exciting story has developed over the last six years through a number of high-impact publications, and this is in part down to the work of the authors of this review, who have played a major role in initiating this field. Their well-written and timely review provides an introduction to iRhoms and related proteins, describes their role in protein turnover and trafficking of client proteins (including ADAM17 and others), mechanisms of iRhom function, roles in disease, and a thoughtful conclusion that nicely emphasises the current knowledge gaps and areas of future research priority. The five figures and two tables are helpful inclusions.

My specific comments are as follows.

1. The current lack of clarity concerning the phenotype of iRhom1 knockout mice is mentioned a couple of times in the review. It would be useful if the authors could comment on why they think that the two publications on this show different phenotypes, particularly as they are responsible for one of the papers.
2. It would be useful if an extra sentence or two could be added to explain the mechanisms of action of the two iRhom client proteins STING and VISA. At present, their introductions are somewhat vague.

3. On a couple of occasions, the relative conservation between regions of iRhom1 and 2 are described, but using rather vague terms such as “least conserved” and “most highly conserved”. It would be useful if the authors could add the precise percentage amino acid identities of the different regions shown on Figure 4, by comparing the human iRhom1 and 2 sequences.

4. In Table 1, it would avoid confusion if “TACE” was changed to “ADAM17”, because the latter is used in the main text.

5. In Table 2, the arthritis reference 14 should be changed to 62. Also, as a minor point, the breast cancer and neurological disease rows do not really need “(i)”, because there is only one phenotype listed.

Decision letter (RSOB-19-0003.R0)

28-Jan-2019

Dear Professor Freeman,

We are pleased to inform you that your manuscript RSOB-19-0003 entitled "The molecular, cellular and pathophysiological roles of iRhom pseudoproteases" has been accepted by the Editor for publication in Open Biology. The reviewer(s) have recommended publication, but also suggest some minor revisions to your manuscript. Therefore, we invite you to respond to the reviewer(s)' comments and revise your manuscript.

Please submit the revised version of your manuscript within 14 days. If you do not think you will be able to meet this date please let us know immediately and we can extend this deadline for you.

1) A text file of the manuscript (doc, txt, rtf or tex), including the references, tables (including captions) and figure captions. Please remove any tracked changes from the text before submission. PDF files are not an accepted format for the "Main Document".

2) A separate electronic file of each figure (tiff, EPS or print-quality PDF preferred). The format should be produced directly from original creation package, or original software format. Please note that PowerPoint files are not accepted.

3) Electronic supplementary material: this should be contained in a separate file from the main text and meet our ESM criteria (see <http://royalsocietypublishing.org/instructions-authors#question5>). All supplementary materials accompanying an accepted article will be treated as in their final form. They will be published alongside the paper on the journal website and posted on the online figshare repository. Files on figshare will be made available approximately one week before the accompanying article so that the supplementary material can be attributed a unique DOI.

Online supplementary material will also carry the title and description provided during submission, so please ensure these are accurate and informative. Note that the Royal Society will not edit or typeset supplementary material and it will be hosted as provided. Please ensure that the supplementary material includes the paper details (authors, title, journal name, article DOI). Your article DOI will be 10.1098/rsob.2016[last 4 digits of e.g. 10.1098/rsob.20160049].

4) A media summary: a short non-technical summary (up to 100 words) of the key findings/importance of your manuscript. Please try to write in simple English, avoid jargon, explain the importance of the topic, outline the main implications and describe why this topic is newsworthy.

Images

Data-Sharing

It is a condition of publication that data supporting your paper are made available. Data should be made available either in the electronic supplementary material or through an appropriate repository. Details of how to access data should be included in your paper. Please see <http://royalsocietypublishing.org/site/authors/policy.xhtml#question6> for more details.

Sincerely,

The Open Biology Team
<mailto:openbiology@royalsociety.org>

Reviewer(s)' Comments to Author:

Referee: 1

Comments to the Author(s)

The review article by Dulloo et al reviews the current knowledge of the molecular, cellular and pathophysiological role of iRhom pseudoproteases. The group of Prof. Freeman has been instrumental in uncovering the various activities of iRhom proteins and the authors did a magnificent job in summarizing these findings of the past 20 years. iRhoms are crucially needed

for the trafficking and activation of the important ADAM17 protease, which governs at least three important pathways, namely the TNF α , IL-6 and EGF-R signaling pathways. These pathways are involved in the regulation of inflammation, infection, metabolism and cancer. In addition, there are additional clients of iRhoms, which are not very well studied yet, and which will possibly change the direction of the field considerably. All important aspects of iRhom biology are nicely covered in the review.

The review article is very well written and it will appear timely since more and more aspects of inflammation and cancer research seem to involve iRhom activities.

There are some points, the authors might want to consider.

1. In the abstract, the authors write that the function of ADAM17 is shedding of cytokines and growth factors. ADAM17 is also an important sheddase of many receptors and adhesion proteins. This could be mentioned here.
2. On p3, the authors describe the different phenotypes of iRhom1 knock-out mice. Do the authors have a possible explanation for the observed variation on phenotypes?
3. On p5, the authors describe the role of iRhom2 in the trafficking of ADAM17 from the ER to the Golgi, where ADAM17 is processed by furin proteases. In contrast to the statement of the authors, this furin cleavage does not lead to activation of ADAM17. This statement should be rephrased.
4. On p12, the authors describe the transcriptional activation of iRhoms. They might want to include a short statement on the structure of the promoters of the iRhom genes, which might illustrate which pathways are likely to affect transcription of iRhom genes.
5. Some references (20, 21, 33, 39, 65, 67, 70) are incomplete and should be amended.
6. In the inset of Fig. 1, the authors might want to indicate the iRhom homology domain (IRHD).
7. As far as this reviewer understands, the data summarized in Fig. 5 are not from Cavadas et al but rather from Oikonomidi et al. This should be corrected.

Referee: 2

Comments to the Author(s)

The iRhom pseudoproteases are emerging as important regulators of the metalloprotease ADAM17, which has a critical role in the body's first line of defence against infection (via shedding of TNF α) and repair (via shedding of EGF receptor ligands). This exciting story has developed over the last six years through a number of high-impact publications, and this is in part down to the work of the authors of this review, who have played a major role in initiating this field. Their well-written and timely review provides an introduction to iRhoms and related proteins, describes their role in protein turnover and trafficking of client proteins (including ADAM17 and others), mechanisms of iRhom function, roles in disease, and a thoughtful conclusion that nicely emphasises the current knowledge gaps and areas of future research priority. The five figures and two tables are helpful inclusions.

My specific comments are as follows.

1. The current lack of clarity concerning the phenotype of iRhom1 knockout mice is mentioned a couple of times in the review. It would be useful if the authors could comment on why they think that the two publications on this show different phenotypes, particularly as they are responsible for one of the papers.

2. It would be useful if an extra sentence or two could be added to explain the mechanisms of action of the two iRhom client proteins STING and VISA. At present, their introductions are somewhat vague.

3. On a couple of occasions, the relative conservation between regions of iRhom1 and 2 are described, but using rather vague terms such as “least conserved” and “most highly conserved”. It would be useful if the authors could add the precise percentage amino acid identities of the different regions shown on Figure 4, by comparing the human iRhom1 and 2 sequences.

4. In Table 1, it would avoid confusion if “TACE” was changed to “ADAM17”, because the latter is used in the main text.

5. In Table 2, the arthritis reference 14 should be changed to 62. Also, as a minor point, the breast cancer and neurological disease rows do not really need “(i)”, because there is only one phenotype listed.

Author's Response to Decision Letter for (RSOB-19-0003.R0)

See Appendix A.

Decision letter (RSOB-19-0003.R1)

19-Feb-2019

Dear Professor Freeman,

We are pleased to inform you that your manuscript entitled "The molecular, cellular and pathophysiological roles of iRhom pseudoproteases" has been accepted by the Editor for publication in Open Biology.

Sincerely,

The Open Biology Team
mailto: openbiology@royalsociety.org

Appendix A

Dulloo et al. Response to Referees

Referee: 1

1. In the abstract, the authors write that the function of ADAM17 is shedding of cytokines and growth factors. ADAM17 is also an important sheddase of many receptors and adhesion proteins. This could be mentioned here.

- It is true that ADAM17 has multiple other roles but in the context of iRhom regulation, it is only cytokines and growth factors that have been reported. It therefore feels appropriate in the abstract of this review to retain that focus.

2. On p3, the authors describe the different phenotypes of iRhom1 knock-out mice. Do the authors have a possible explanation for the observed variation on phenotypes?

- We believe the differences could be due to the way the genetic deletion of iRhom1 was done in each mice. In Christova et al., exons 2-18 of iRhom1 were deleted, encompassing the whole coding sequence of the gene. In Li et al., only exons 4-11 were removed, which does not exclude the possibility of a shorter form of iRhom1 still being expressed. Several shorter transcripts of iRhom1 have been described on UCSD genome browser to exist due to alternative splicing events as highlighted in the text on Page 12. Also, the different in genetic background of the mice used can also be a contributing factor. A statement has been included in the text.

3. On p5, the authors describe the role on iRhom2 in the trafficking of ADAM17 from the ER to the Golgi, where ADAM17 is processed by furin proteases. In contrast to the statement of the authors, this furin cleavage does not lead to activation of ADAM17. This statement should be rephrased.

- Statement has been modified accordingly.

4. On p12, the authors describe the transcriptional activation of iRhoms. They might want to include a short statement on the structure of the promoters of the iRhom genes, which might illustrate which pathways are likely to affect transcription of iRhom genes.

- A brief description has been made accordingly.

5. Some references (20, 21, 33, 39, 65, 67, 70) are incomplete and should be amended.

-Amendments made accordingly

6. In the inset of Fig. 1, the authors might want to indicate the iRhom homology domain (IRHD).

- Label added accordingly

7. As far as this reviewer understands, the data summarized in Fig. 5 are not from Cavadas et al but rather from Oikonomidi et al. This should be corrected.

- Actually, we are right! The data summarised is taken from Cavadas et al., (main figures and the supplementary figures of the paper which details the indicated interactors of iRhom2).

Referee: 2

1. The current lack of clarity concerning the phenotype of iRhom1 knockout mice is mentioned a couple of times in the review. It would be useful if the authors could comment on why they think that the two publications on this show different phenotypes, particularly as they are responsible for one of the papers.

- We believe the differences could be due to the way the genetic deletion of iRhom1 was done in each mice. In Christova et al., exons 2-18 of iRhom1 were deleted, encompassing the whole coding sequence of the gene. In Li et al., only exons 4-11 were removed, which does not exclude the possibility of a shorter form of iRhom1 still being expressed. Several shorter transcripts of iRhom1 have been described on UCSD genome browser to exist due to alternative splicing events as highlighted in the text on Page 12. Also, the different in genetic background of the mice used can also be a contributing factor. A statement has been included in the text.

2. It would be useful if an extra sentence or two could be added to explain the

mechanisms of action of the two iRhom client proteins STING and VISA. At present, their introductions are somewhat vague.

- Further statements about the role of STING and VISA have been included in the text

3. On a couple of occasions, the relative conservation between regions of iRhom1 and 2 are described but using rather vague terms such as “least conserved” and “most highly conserved”. It would be useful if the authors could add the precise percentage amino acid identities of the different regions shown on Figure 4, by comparing the human iRhom1 and 2 sequences.

- Statements about percentage conservancy of amino acids for indicated domains between iR1 and iR2 have been included in text.

4. In Table 1, it would avoid confusion if “TACE” was changed to “ADAM17”, because the latter is used in the main text.

- TACE has been changed to ADAM17 accordingly

5. In Table 2, the arthritis reference 14 should be changed to 62. Also, as a minor point, the breast cancer and neurological disease rows do not really need “(i)”, because there is only one phenotype listed.

- Changes have been made accordingly.